# Pharmacological and Nutritional Modulation of Vascular Calcification

**DOI:** 10.3390/nu12010100

**Published:** 2019-12-30

**Authors:** Liv M. Vossen, Abraham A. Kroon, Leon J. Schurgers, Peter W. de Leeuw

**Affiliations:** Department of Medicine and Department of Biochemistry, Maastricht University Medical Center and Cardiovascular Research Institute Maastricht (CARIM), P.O. Box 5800, 6202 AZ Maastricht, The Netherlands; liv.vossen@mumc.nl (L.M.V.); aa.kroon@mumc.nl (A.A.K.); l.schurgers@maastrichtuniversity.nl (L.J.S.)

**Keywords:** vascular calcification, matrix Gla protein, vitamin K

## Abstract

Vascular calcification is an independent predictor of cardiovascular disease, and therefore, inhibition or regression of this processes is of clinical importance. The standard care regarding prevention and treatment of cardiovascular disease at this moment mainly depends on drug therapy. In animal and preclinical studies, various forms of cardiovascular drug therapy seem to have a positive effect on vascular calcification. In particular, calcium channel blockers and inhibitors of the renin–angiotensin–aldosteron system slowed down arterial calcification in experimental animals. In humans, the results of trials with these drugs are far less pronounced and often contradictory. There is limited evidence that the development of new atherosclerotic lesions may be retarded in patients with coronary artery disease, but existing lesions can hardly be influenced. Although statin therapy has a proven role in the prevention and treatment of cardiovascular morbidity and mortality, it is associated with both regression and acceleration of the vascular calcification process. Recently, nutritional supplements have been recognized as a potential tool to reduce calcification. This is particularly true for vitamin K, which acts as an inhibitor of vascular calcification. In addition to vitamin K, other dietary supplements may also modulate vascular function. In this narrative review, we discuss the current state of knowledge regarding the pharmacological and nutritional possibilities to prevent the development and progression of vascular calcification.

## 1. Introduction

Arterial calcification is a strong and independent predictor of cardiovascular morbidity and mortality [1,2,3,4]. Therefore, regression or inhibition of calcification is of clinical importance. Presently, the standard care regarding prevention and treatment of cardiovascular disease depends mainly on drug therapy [5]. However, while preclinical and animal studies have shown that in particular calcium channel blockers (CCBs) and inhibitors of the renin–angiotensin system have favorable effects on vascular calcification [6,7,8,9,10], the results of studies with, e.g., CCBs in humans are far less impressive and often difficult to interpret [11,12,13,14]. Although statin therapy also has a proven role in the prevention and treatment of cardiovascular morbidity and mortality [15], it does not materially affect the rate of progression of coronary calcification [16]. More recently, even an accelerated increase in coronary artery calcification (CAC) was seen during statin treatment [17,18]. Altogether, the effects of conventional drug therapy on vascular calcification seem to be a bit disappointing. This has prompted several investigators to search for alternative methods to slow down the vascular calcification process. In this regard, dietary interventions with certain vitamins, notably vitamin K, have yielded promising results [19]. In addition to vitamin K, other dietary supplements (vitamin B, C, D, E, electrolytes, antioxidants) have been tested for their potential to modulate vascular function. One should bear in mind, though, that vascular calcification takes a long time to develop and that it is difficult, therefore, to study the effects of nutritional treatment on this process. 

Our aim is to present here a critical review of studies, both laboratory and clinical, which have examined the effects of pharmacological and nutritional interventions on the development or progression of vascular calcification. 

## 2. Search Strategy

For this narrative review of the literature, we explored PubMed, the Cochrane Library, and EMBASE up to 1 November 2019 using the following search terms: ‘vascular calcification’ or ‘arterial calcification’ or ‘coronary artery calcification’; for preclinical and animal data we added ‘vascular smooth muscle cell’. These search terms were used in any combination with keywords for drug therapy (‘calcium channel blocker’, ‘renin-angiotensin-aldosterone blocker’, ‘angiotensin converting enzyme inhibitor’, ‘angiotensin-receptor blockers’, ‘statin’), vitamin K supplementation (‘vitamin K’, ‘menaquinone’, ‘menaquinone-7′, ‘vitamin K2′, ‘vitamin K supplementation’) and dietary supplements (‘vitamin B’, ‘vitamin C’, ‘ascorbic acid’, ‘calcium supplements’, ‘vitamin D’, ‘vitamin supplementation’, ‘vitamin E’ and magnesium’).

The search was limited to full text papers, clinical trials, observational studies, and reviews in English language and resulted in a total number of 3309 hits in Pubmed, 1083 hits in the Cochrane Library, and 5587 hits in EMBASE. By screening titles and abstracts, 138 articles were considered to be eligible for inclusion in our review. Reference lists of included articles and appropriate reviews were screened for additional studies. This resulted in 4 additional papers. When multiple papers with similar data from the same research group were available, we used only the publication with the largest population. In addition, when papers had been included in systematic reviews or meta-analyses, we only used the aggregate results. For a discussion of the final data that we retrieved, we divided the papers into those dealing with pharmacological treatment and those addressing nutritional support.

## 3. Pathophysiological Aspects of Vascular Calcification

Although it is beyond the scope of this paper to discuss in-depth the mechanisms that are involved in vascular calcification, we briefly touch here upon the most important pathophysiological pathways so that the rationale of some treatments can be better understood. 

Under normal circumstances, contractile vascular smooth muscle cells (VSMCs) which are able to take up calcium through calcium channels in their membrane regulate vessel wall tone and synthesize the calcification inhibitor matrix Gla-protein (MGP), which makes them resistant to calcification. Before being biologically active, MGP requires posttranslational carboxylation of specific protein bound glutamate-residues, a process which is catalyzed by the vitamin K dependent enzyme gamma-glutamylcarboxylase [20]. A variety of stress signals (Table 1) can induce a phenotypic switch of VSMCs towards an osteoblast-like cell type which contributes to pathological vascular remodeling in both the media and the intima. To prevent apoptosis or calcification, VSMCs produce extracellular vesicles which are loaded with carboxylated, and hence active, MGP to prevent calcification nucleation [21]. In case of ongoing calcification pressure or, for instance, vitamin K deficiency, the extracellular vesicles may be faced with an excess of the inactive, uncarboxylated MGP, which makes them prone to support calcification. Once the extracellular matrix is calcified, VSMCs may differentiate into osteochrondrogenic VSMCs which produce less MGP, synthesize bone-associated proteins, and further fuel the vascular mineralization process, thus causing a vicious cycle [20]. Contrary to what was previously thought, calcification is not a passive phenomenon but a highly controlled process which involves many regulatory proteins. 

Medial calcification, which presents as rail-tracking deposits along the vasculature, is associated with aging but is particularly prevalent in patients suffering from chronic kidney disease (CKD) and diabetes mellitus. It preferentially affects the lower aorta and peripheral arteries, e.g., of the limbs. In patients with CKD, a disturbance in calcium–phosphate balance, accumulation of uremic toxins, severe vitamin K deficiency, and a lack of calcification inhibitors have been implicated in the pathogenesis of calcification [22]. In type 2 diabetes, several signaling pathways such as elevated glucose and insulin levels may be operative, but the precise mechanisms are still not fully understood [23]. In view of the key role played by vitamin K, it is not surprising that patients with vitamin K deficiency and those who are using long-term anticoagulant therapy with vitamin K antagonists are also prone to develop medial calcification [24]. 

Intimal calcification shows a much patchier distribution pattern than medial calcification, commonly affects the carotid and coronary arteries, and is typically related to atherosclerosis, which is considered an inflammatory disease with endothelial damage due to, e.g., dyslipidemia, hypertension, and smoking. It is not uncommon, though, to find both types of calcification in the same patient. For more detailed information, the reader is referred to some recent reviews on this topic [3,23,25,26,27]. Although it is possible to measure many of the proteins that are involved in the regulation of calcification, there are as yet no established biomarkers available that could help the clinician to monitor a patient’s calcification status. 

## 4. Pharmacological Treatment

Although there is a vast array of available cardiovascular drugs, data regarding their effects on vascular calcification are available for only a few of them, notably calcium channel blockers, inhibitors of the renin–angiotensin system, and statins. Other drugs that have been tested for their potential calcification-inhibiting properties are bisphosphonates and denosumab. Given that inflammatory processes in the vascular wall form an integral part of atherosclerosis and calcification, anti-inflammation drugs could theoretically be useful to reduce calcification. However, with the exception of statins, which have anti-inflammatory properties, there are no clinical studies with specific anti-inflammatory agents.

### 4.1. Calcium Channel Blockers

Calcium channel blockers are drugs which inhibit voltage-dependent calcium channels in the plasma membrane and thereby diminish intracellular calcium content. In VSMCs, this may lead to a reduced calcium content of extracellular vesicles and thus lead to less calcification of the vascular wall. Indeed, in vivo animal studies suggest that CCBs block vascular calcification in coronary arteries of rats with calcium-induced arteriosclerosis, spontaneously hypertensive rats, and cholesterol-fed rabbits [6,8,10]. In bovine VSMCs, the CCB verapamil appeared to significantly decrease calcification of these cells in a dose-dependent manner but by a mechanism independent of the CCB effect [28]. Another study which used human aortic VSMCs in vitro found that the CCB azelnidipine inhibited osteogenic differentiation and matrix mineralization of these cells [29]. However, despite these seemingly promising data, the outcome of clinical trials is less clear. The first study to address the possible effect of a CCB on coronary calcification was the International Nifedipine Trial on Anti-atherosclerotic Therapy (INTACT) trial which was a prospective, placebo-controlled, randomized, double-blind multicenter trial to investigate the effect of the CCB nifedipine on the progression of coronary artery disease [30]. This study did, indeed, show a significant reduction in the formation of new coronary lesions in patients with coronary artery disease who were treated with nifedipine. However, no effect was seen on the progression of existing lesions. Moreover, the method used to quantify coronary calcification was not very sensitive. Comparable studies using either amlodipine [9] or nicardipine [31] also failed to show any effect whatsoever of these drugs on established coronary lesions beyond that of what could be explained by blood pressure control alone. 

As part of the International Nifedipine GITS study: Intervention as a Goal in Hypertension Treatment (INSIGHT) trial, Motro and coworkers applied double-helix computerized tomography in a subset of the patients to assess coronary calcification [13]. The reason to use this imaging modality was that with this technique, it became possible to directly estimate the degree of calcification of the coronary arteries. The study population consisted of hypertensive patients with a high-risk profile who were treated with either nifedipine or co-amilozide [32]. Although nifedipine leads to regression of coronary calcification when compared to that of co-amilozide, the intention-to-treat analysis did not show a significant difference. In addition, the main results of the trial showed equal effectiveness in the prevention of cardiovascular complications. Thus, there exists little evidence, if at all, that CCBs are able to significantly modify progression of coronary artery calcification [3]. As far as calcification of peripheral arteries is concerned, evidence for a positive effect of CCBs is also lacking. 

### 4.2. Renin–Angiotensin System Inhibition

Experimental studies indicate that drugs interfering with the formation or the function of the pro-atherogenic angiotensin II (Ang II) have a beneficial effect on vascular stiffness and calcification. It is thought that this effect is related to the ability of these drugs to improve endothelial function by preventing VSMC proliferation and migration as well as extracellular matrix remodeling [7,33]. Via activation of receptor activator of nuclear factor [kappa]B ligand (RANKL), Ang II was found to increase in vitro calcium deposition in human aortic VSMCs and to promote the phenotypic switch of these cells into osteoblast-like VSMCs [34]. Pharmacological blockade with the angiotensin receptor blocker (ARB) olmesartan was able to block this process. Olmesartan also inhibited vascular calcification induced by an atherogenic diet in normotensive rabbits [7]. Angiotensin-(1–7), the naturally occurring antagonist of Ang II, inhibits vascular calcification by retarding the osteogenic transition of VSMCs in rats [35], and several other experimental studies have shown that both ARBs and inhibitors of angiotensin converting enzyme (ACE) can effectively suppress vascular calcification and stiffness [36,37,38]. Finally, in vitro studies using human mesenchymal stem cells showed that blockade of the Ang II type 2 receptor inhibited osteogenic differentiation, indicating that the Ang II type 2 receptor is also involved in calcification [39]. 

Notwithstanding these experimental data, results of clinical trials in patients with coronary disease are less clear. In the Comparison of Amlodipine vs. Enalapril to Limit Occurrences of Thrombosis (CAMELOT) study, for instance, the effect of the ACE-inhibitor enalapril on progression of atherosclerosis was nonsignificant and less pronounced than that of the CCB amlodipine [40]. In another study, the Perindopril’s Prospective Effect on Coronary atherosclerosis by Angiography and Intravascular Ultrasound Evaluation (PERSPECTIVE) trial, vessel, lumen, and plaque areas were measured by intracoronary ultrasound during treatment with either placebo or perindopril. Although plaques with no or little calcification did show some regression on perindopril, the ACE-inhibitor had no measurable effect on plaques with moderate or severe amounts of calcification [41]. Still, several other trials, such as the Heart Outcomes Prevention Evaluation (HOPE) study, the European trial of Reduction Of cardiac events with Perindopril in stable coronary Artery disease (EUROPA) study, and the Ongoing Telmisartan Alone and in Combination With Ramipril Global Endpoint Trial (ONTARGET), showed beneficial effects of blockade of the renin–angiotensin system on cardiovascular events [42,43,44]. 

### 4.3. Statins

Statins lower low-density lipoprotein (LDL)-cholesterol and reduce inflammation in the vascular wall. Thus, they potentially reduce some important triggers for the osteogenic differentiation of VSMCs. However, studies on the effect of statins on the calcification process show somewhat erratic results. For instance, in a model of inflammatory vascular calcification, statins inhibited calcification of human VSMCs in a dose-dependent manner [45], while atorvastatin dose-dependently stimulated calcification when VSMCs of aortas explanted from Wistar rats were incubated in a special calcification medium [46]. Atorvastatin also induced considerable apoptosis of the VSMCs. 

Clinical studies also show disparate results. Indeed, several observational and prospective studies suggested that lipid-lowering therapy with statins decreases coronary vascular calcification or, at least, slows down the progression of calcification [47,48,49]. Other studies, however, could not confirm this trend and showed either no effect [50,51,52,53] or even an increase in calcium scores during treatment [54,55,56]. Changes in calcium scores appear to be unrelated to the degree of LDL-cholesterol lowering. On the basis of a meta-analysis of five controlled trials that assessed the effect of statins on CAC and six trials that looked at coronary stenoses, Henein and colleagues concluded that LDL-cholesterol-lowering therapy with statins is associated with a reduction of noncalcified coronary atherosclerotic plaque burden, but that it does not affect the rate of progression of coronary calcification [16]. When the same group later reanalyzed the data from two other trials, they found that statin treatment increased CAC as compared to placebo. However, there were no more events in statin-treated patients. This suggests that CAC growth under statin treatment represents plaque stabilization rather than plaque expansion [17]. Finally, Puri and coworkers performed a post-hoc patient-level analysis of eight prospective randomized trials in which changes of coronary atheroma volume and calcium indices were measured. This analysis also showed that statins promote coronary atheroma calcification independently of their plaque-regressive effects [18]. Recently, Lee and coworkers reported on the results of the Progression of Atherosclerotic Plaque Determined by Computed Tomographic Angiography Imaging (PARADIGM) study, which is a dynamic, multinational observational registry that prospectively collects data of patients who have undergone serial coronary computed tomography angiography [57]. Their data show that without statin therapy, any increase in the coronary calcium score reflects progression in both previously noncalcified and already calcified plaque volumes. However, when statins are given, an increase in calcium score indicates only progression in calcified plaques. 

### 4.4. Bisphosphonates and Denosumab

Given the similarities of the molecular mechanisms that are involved in vascular calcification and bone mineralization, the idea that drugs like bisphosphonates and denosumab may have an anti-atherosclerotic or anticalcification effect is not too far-stretched. Bisphosphonates are not only accumulating in bone but also in (calcified) atherosclerotic plaques, and in macrophages, they inhibit the cellular accumulation of LDL-cholesterol [58]. Rosenblum and colleagues were among the first to describe that the bisphosphonate etidronate suppresses the formation of atherosclerotic lesions and in a dose-related manner reduced calcification in a rabbit model of athero-arteriosclerosis [59]. Since then, several studies in other animal models have obtained comparable results, primarily with nitrogen-containing bisphosphonates. Initial studies in humans were carried out in patients on long-term hemodialysis and confirmed that bisphosphonates might limit vascular calcification [60,61]. Subsequently, the Multi-Ethnic Study of Atherosclerosis (MESA) demonstrated in a large group of elderly women of various ethnic backgrounds that the use of nitrogen-containing bisphosphonates is associated with a decreased prevalence of cardiovascular calcification [62]. However, most human studies with bisphosphonates in vascular disease suffer from small sample sizes, a short duration of treatment, and the selection of patients with other risk factors that could have confounded the results. Although bisphosphonates may, indeed, have an inhibitory effect on atherosclerosis, their effect is variable, small, and probably not of great clinical importance [63]. 

The humanized monoclonal antibody denosumab is another antiresorptive agent that binds to RANKL and attenuates vascular calcium deposition in experimental animals [64]. So far, however, there is no evidence that it has a major effect on vascular calcification in humans [65]. 

## 5. Nutritional Supplements

With respect to the nutritional elements that have been applied to try to reduce or retard vascular calcification, most attention has been given to vitamin K. Nevertheless, other vitamins (A, B, C, D, and E), electrolytes (calcium, magnesium, and phosphate) and some other substances, e.g., garlic, need to be briefly discussed as well. A summary of what is known is presented in Table 2. 

### 5.1. Vitamin K Supplementation

Vitamin K consists of a group of structurally related compounds that include phylloquinone (vitamin K1) and menaquinones (vitamin K2). Phylloquinone still is the most used synthetic nutritional supplement worldwide, but in recent years, other vitamin K supplements were introduced which predominantly contain menaquinone-4 (MK-4) and menaquinone-7 (MK-7). The latter is the most hydrophobic vitamin K with good bioavailability and a long half-life [66]. The rationale for vitamin K to promote vascular health lies in its function as an essential cofactor in the activation of several extracellular matrix proteins, notably MGP, that inhibit vascular calcification. 

Schurgers and coworkers were the first to demonstrate in rats that dietary supplementation with either phylloquinone or MK-4 markedly attenuated the development of warfarin-induced medial vascular calcification but only when high doses were administered [66]. This study also confirmed the crucial role of MGP in inhibiting calcification. A later study in rats with adenine-induced chronic renal failure showed that also in this model of chronic kidney disease, administration of a high dose of vitamin K lessened the development of warfarin-induced calcification [67]. There is evidence to show that at the tissue level, vitamin K1 is converted to menaquinone-4 [68], so it may be that dietary supplementation with vitamin K2 or its analogues may be preferable above administration of high doses of vitamin K1. 

Observational data in humans are in line with the potential benefit of vitamin K2 on CAC and vascular stiffness. The population-based Rotterdam study, for instance, showed an inverse relationship between menaquinone intake and abdominal aortic calcification, while no such relationship was apparent for phylloquinone [69]. These findings were corroborated by Beulens and coworkers, who found in postmenopausal women that a high intake of menaquinone but not of phylloquinone was associated with reduced coronary calcification as assessed by multidetector-computed tomography [70]. In line with these data, a double-blind randomized controlled trial where equal numbers of older men and women, free of clinically manifest cardiovascular disease, were assigned to receive a daily multivitamin preparation with or without phylloquinone showed that this vitamin K supplement did not significantly slow down the progression of CAC over a 3-year follow-up period [71]. However, while this was true in the intention-to-treat analysis, those participants in whom coronary artery calcification was present before treatment exhibited less progression of calcification than their counterparts without pre-existing lesions. The data further suggest that phylloquinone may be able to somewhat alter the course of existent lesions, but that it does not reduce the development of new vascular calcifications. More recently, one-year vitamin K1 supplementation in patients with aortic valve calcification showed a 50% reduction of progression of calcification compared to placebo [72]. 

Several studies including some randomized trials have reported an improvement in vitamin K status and the degree of MGP carboxylation with supplementation of menaquinone [73]. Although MK-7 supplements may improve arterial stiffness [74], data from various trials are not consistent. Recently, Lees and coworkers performed a meta-analysis of clinical trials that assessed the effect of vitamin K (1 or 2) supplementation on vascular function and found that compared to control measures, vitamin K supplementation significantly reduced the progression of vascular calcification [19]. However, this conclusion was based on only three trials, two of which evaluated vitamin K1 and only one vitamin K2. 

### 5.2. Supplementation of Other Vitamins

Little is known about the effects of vitamin B on vascular function. A recent study from Brazil in patients with chronic kidney disease suggested that a higher intake of pantothenic acid (vitamin B5) may have a small protective effect with respect to calcification, but this seems hardly to be of clinical significance [75]. Another recent study measured pulse wave velocity as a proxy for vascular calcification but failed to find an association between dietary intake of B-vitamins and vascular function [76]. Further, for vitamin C supplementation, there is currently no evidence that this has any clinical meaningful effect on vascular function [77]. Finally, there is no evidence that supplements of vitamin D or E have any positive effect on the vasculature [78,79]. 

As far as the effects of vitamin A are concerned, there are no data in humans that have demonstrated a relationship between vitamin A status and vascular calcification, nor are there any clinical trials in this regard [80]. There is even information from studies in mice that increased dietary intake of vitamin A may promote heart valve calcification [81].

### 5.3. Supplementation or Restriction of Electrolytes

Although calcium plays a key role in the calcification process, the extracellular concentration of this electrolyte is so high relative to its intracellular concentration that changes in dietary calcium are unlikely to have a major effect on vascular function, at least in individuals without renal insufficiency. Indeed, neither a positive nor an adverse effect of dietary calcium or calcium supplements on arterial calcification has been found in observational studies, small-scale trials, and meta-analyses [82,83,84,85,86]. Currently, there is also too little supportive evidence for any advantage of dietary phosphate restriction [87]. 

With respect to magnesium, available data are more promising. Indeed, various observational studies show that a higher magnesium intake is associated with less vascular calcification, and small-scale intervention trials suggest that magnesium supplementation may slow the progression of CAC [88].

### 5.4. Antioxidants

Recently, attention has been given to some other dietary measures that could have a bearing on the calcification process. For instance, antioxidants, other than the vitamins mentioned above, are becoming increasingly popular as a means to promote vascular health. There is, however, hardly any scientific information to support such a view. Although flavanoids, polyphenols, and alpha-lipoic acid may have antioxidant properties, there is no single study to show a positive effect of these substances on vascular calcification in humans [80]. Perhaps the results obtained with aged garlic extract which also acts as an antioxidant, stand out as an exception to these negative findings. In a recent, randomized placebo-controlled trial, aged garlic extract in combination with vitamin supplements significantly reduced markers of coronary calcification after one year of treatment in patients with a coronary calcium score of more than 30% at baseline [89]. It should be noted, though, that all participants were also taking a statin and a low-cholesterol diet and that the degree of calcification was not measured directly. A meta-analysis of three trials also noted the anticalcifying effect of garlic, but both the meta-analysis and the three individual trials came from the same research group [90], and there has been no independent confirmation. 

### 5.5. Other Food Constituents

Unsaturated fatty acids, carbohydrates, and proteins could all be expected to play a role in the modulation of vascular or heart valve calcification, but so far, there are no solid data to support dietary adaptation of these food constituents [81]. 

## 6. Discussion

At present, there is general agreement that vascular calcification is not a passive phenomenon but a highly regulated process [25]. This renders it potentially amenable to treatment. However, the mechanisms involved in the process are still not fully understood. The somewhat naive thought that we would be able to reduce calcium deposition in atherosclerotic plaques by modulation of dietary calcium certainly is not borne out by facts. Observational studies have failed to show a relationship between calcium intake and vascular calcification, and supplemental calcium in the diet, once thought to protect against cardiovascular complications, does not have a measurable effect either. This suggests that vascular calcification is not simply a matter of calcium excess. Alternatively, cellular calcium influx could theoretically be important as the osteogenic differentiation and mineralization of VSMCs is an essential step in the pathogenesis of vascular calcification. 

Since CCBs can block the conversion and calcification of VSMCs, it is surprising that so little clinical information is available with respect to the effects of CCBs on calcification. Inasmuch as there are data from clinical trials, this shows that CCBs do not prevent progression of existing plaques, but they may be able to slow down the development of new lesions, although the evidence for such an effect is still very slim and not enough to advocate CCBs for that purpose. This suggests that only blocking calcium channels is not enough to prevent or retard calcification. Alternatively, it may be that these drugs are unable to reach the sites where active mineralization takes place. Inasmuch as CCBs are able to retard the development of new lesions, this may well be related to their ability to improve hemodynamics and not so much to a direct effect on calcification. 

Notwithstanding the preventive effects of inhibitors of the RAAS on cardiovascular events, and in spite of their calcification-inhibiting effects under experimental conditions, there is no convincing evidence that these drugs can inhibit or reverse calcification in patients with cardiovascular disease. It is likely that the positive effects of inhibitors of the renin–angiotensin system with respect to cardiovascular complications are primarily related to their vasodilating properties and not to modification of calcification. Again, the question is also whether these drugs are able to penetrate in those tissue segments where calcification takes place. 

Bisphosphonates, by contrast, seem a bit more promising, but the current evidence for their ability to inhibit calcification is fairly limited. Nevertheless, it would be worthwhile to set up larger trials to evaluate whether this type of drugs could be useful as an adjunct to anti-atherosclerotic treatment. 

In fact, the only drugs that seem to have a clinically measurable effect on calcification are statins. Interestingly, these drugs act as a two-sided sword. On the one hand, they lower LDL-cholesterol and reduce atheroma volume. On the other, they promote plaque calcification and via that mechanism may increase plaque stability [18]. 

Taken together, the available data suggest that we are as yet unable to reverse plaque calcification with drug treatment, at least in advanced stages of the atherosclerotic process. It is still possible that the pharmacological approach is helpful in the early phases of the disease when most patients will still be asymptomatic, but such patients normally will not come to the attention of the physician. 

The question is whether we can prevent the development or the progression of atherosclerotic lesions and calcification by lifestyle changes, such as stopping smoking, increasing exercise, and dietary measures. Although such measures probably help to reduce calcification, there is no compelling scientific evidence that they really do so (Table 2). Admittedly, recent studies have provided some evidence that a favorable lifestyle, including a healthy diet, may substantially reduce the risk of coronary artery disease compared to an unhealthy lifestyle [91,92], but these studies have focused more on the overall risk of coronary complications and less, or not at all, on calcification per se. Moreover, a healthy diet is usually defined in broad terms (amount of vegetables, fruits, etcetera) without paying attention to specific food constituents. Thus, more detailed information is needed about how certain food components could modify the calcification process. As indicated above, there seems to be no role for calcium intake in this regard, but preliminary data obtained with supplemental magnesium look promising. Magnesium is known to inhibit the calcification of VSMCs and to upregulate the expression of MGP, making this electrolyte a true calcification inhibitor [88]. So far, however, the effect of magnesium has only been studied in a small group of hemodialysis patients and patients with preterminal chronic kidney disease. 

As far as the administration of vitamins is concerned, it seems useless to take supplemental doses of the vitamins A, B, C, D, and E as part of an anticalcification strategy. However, a promising candidate in this regard would be vitamin K. Of all the nutritional supplements, this vitamin presently has the best perspective to become recommended for the prevention or progression of calcification. The biological rationale for a role of this vitamin lies in its ability to activate MGP, which is a potent inhibitor of ectopic calcification. Although the meta-analysis by Lees and coworkers illustrates that there is still a considerable gap in our knowledge [19], the results of controlled trials look promising. Whether high doses of phylloquinone or of one of the menaquinones or both will be best still needs to be explored. A trial investigating specifically the effects of MK-7 supplementation in patients with coronary artery disease is underway and is soon to report its results [93]. Overall, there are still so many uncertainties that the effects of vitamin K on the calcification process are still uncertain. 

Finally, supplementation with antioxidants has no measurable effect on calcification, but perhaps aged garlic extract holds promise. However, data are still limited and in need of confirmation. 

As many, if not most, of the studies on vascular calcification and its modulation by drugs or nutritional supplements have been performed on high-risk populations such as patients on hemodialysis or with predialysis CKD, there is an urgent need to assess whether these interventions are also effective in preventing or slowing vascular abnormalities in individuals with less advanced disease. Large-scale trials with sufficient power will be needed, though, to demonstrate a clinically meaningful outcome. Until the results of such trials become available, the clinician dealing with patients with atherosclerosis may consider prescribing supplemental magnesium and/or vitamin K in an attempt to limit further damage.

## Figures and Tables

**Table 1 nutrients-12-00100-t001:** Factors which may promote vascular calcification.

***Traditional Risk Factors***
Aging
Male sex
Hypertension
Diabetes mellitus
Chronic kidney disease
Dyslipidemia
Smoking
***Stress signals for the vasculature***
Inflammation
Oxidative stress
Shear stress
Advanced glycation products
Increased calcium-phosphate product
High Ang II
ECM degradation
Uremic toxins
Vitamin K deficiency or antagonism

**Table 2 nutrients-12-00100-t002:** Effect of nutritional interventions on vascular calcification.

***Vitamin Supplementation***	
Vitamin A	If anything, progression
Vitamin B	No demonstrable effect
Vitamin C	No demonstrable effect
Vitamin D	No demonstrable effect
Vitamin E	No demonstrable effect
Vitamin K1	Possibly less progression
Vitamin K2	Possibly less progression
***Electrolytes***	
Calcium supplementation	No demonstrable effect
Phosphate restriction	No data
Magnesium supplementation	Possibly less progression
***Others***	
Antioxidants supplementation	No demonstrable effect
Aged garlic extract	Possibly less progression

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
