# Peer review of "Pharmacological and Nutritional Modulation of Vascular Calcification"

_nutrients, 2019, doi:10.3390/nu12010100_

Round 1

Reviewer 1 Report

I would recommend: - Accept - written to a very high standard - Avoid excessive use of abbreviations; only use common abbreviations and if used >10 times - Consider adding a section on nutrition and lifestyle change for reversing coronary artery disease: https://scholar.google.com.au/scholar?q=reversing+coronary+artery+disease+with+diet&hl=en&as_sdt=0&as_vis=1&oi=scholart

Author Response

We thank the reviewers for their comments and suggestions to improve our manuscript.

Responses to reviewer # 1

Avoid excessive use of abbreviations; only use common abbreviations and if used >10 times

Reply

We eliminated nearly all abbreviations to retain only those that are most commonly used.

Consider adding a section on nutrition and lifestyle change for reversing coronary artery disease: https://scholar.google.com.au/scholar?q=reversing+coronary+artery+disease+with+diet&hl=en&as_sdt=0&as_vis=1&oi=scholart

Reply

In the Discussion, we have added a paragraph on the potential effects of lifestyle and nutrition.

Reviewer 2 Report

The article aims to review the literature about drugs used for vascular calcification. Authors were very careful in mention that the text is not about mechanisms, but to review studies (laboratory and clinical) that regarding pharmacological and nutritional interventions against vascular calcification. I understand that in how the text is addressed is part of the authors decision. However, in the current format, the text just informs the treatments and whether they have positive or negative outcomes.

To improve the quality of the information, some changes should be addressed:

The pathophysiology of the disease should be better described. Which group of patients are more in risk of vascular calcification and why? Are there molecular markers that help the clinicians in identify the course of the treatment? I think that some mechanisms of action should be mentioned, as the authors partially in the section 3.2. Some figures or tables should be added. Authors should have a critical point of view in mention why those treatments worked or not. Nutritional data is only tangentially mentioned. A new section about new clinical and nutritional approach should be added.

Author Response

We thank the reviewers for their comments and suggestions to improve our manuscript.

Responses to reviewer # 2

The pathophysiology of the disease should be better described.

Reply

Following the paragraph with the search strategy, we have added a section on pathophysiology.

Which group of patients are more in risk of vascular calcification and why?

Reply

Patients with diabetes mellitus and those with chronic kidney disease are particularly at risk as are the ones who chronically use vitamin K antagonists. We have added this information in the section on pathophysiology.

Are there molecular markers that help the clinicians in identify the course of the treatment?

Reply

Unfortunately, there are no such markers at the present time. We have added a sentence to that effect at the end of the section on pathophysiology. 

I think that some mechanisms of action should be mentioned, as the authors partially in the section 3.2.

Reply

We have added some information on mechanisms of action for the various interventions.

Some figures or tables should be added.

Reply

We have added two tables.

Authors should have a critical point of view in mention why those treatments worked or not.

Reply

In the discussion, we have added a few comments of why certain treatments would or would not work.

Nutritional data is only tangentially mentioned. A new section about new clinical and nutritional approach should be added.

Reply

We have expanded the section on nutritional data.

Reviewer 3 Report

This is a good review.

I have several comments:

Please mention that vascular calcification is a long process. It takes years to develop - hence difficult to study. Vascular calcification in peripheral arteries may differ from calcification in the coronary arteries.

Vascular calcification is often related to diabetes. Please discuss this.

Artery calcification is believed to be related to inflammation. Please discuss use of anti-inflammation agents.

Does stopping smoking help to reduce calcification?

Author Response

We thank the reviewers for their comments and suggestions to improve our manuscript.

Responses to reviewer # 3

Please mention that vascular calcification is a long process. It takes years to develop - hence difficult to study.

Reply

We have added a statement to this effect in the Introduction.

Vascular calcification in peripheral arteries may differ from calcification in the coronary arteries.

Reply

In the section on pathophysiology we have also indicated that there is a difference in medial and intimal calcification which mainly affects peripheral and coronary arteries respectively.

Vascular calcification is often related to diabetes. Please discuss this.

Reply

In the newly added paragraph on pathophysiological aspects, we have mentioned the relationship between diabetes and calcification. 

Artery calcification is believed to be related to inflammation. Please discuss use of anti-inflammation agents.

Reply

We have added a sentence to the manuscript stating that no studies have been done with anti-inflammatory agents (page 6).

Does stopping smoking help to reduce calcification?

Reply

It is likely but the true answer is that we do not know. We have added a sentence to the Discussion to express this uncertainty.

Round 2

Reviewer 2 Report

The review is significantly improved compared the the first version. Thanks for considering my suggestions. I have no additional requirements

Author Response

Thank you very much.

Reviewer 3 Report

The paper is fine other than issues of English/writing:

1. line 105  - a much more patchy....

2. lines 138 - "Moreover, quantitative coronary 138 angiography was used to assess the extent of coronary disease which is not a sensitive method to 139 quantitatively estimate the degree of coronary calcification"  Better to say "However the method used to quantify coronary calcification was not accurate.".

3. Line 150 "there was a positive effect..." You want to say 'nifedipine lead to regression of …".

4. Lines 209 ff  - I do not understand the logic here. Rewrite.

5. Lines 218ff - how can a non calcified area reflect calcification enlargement? This does not make sense.

6. Lines 275-279 - the first half and second half of the sentence do not connect well. Rewrite.

7. Line 299-300 - since the study does not mention calcification, why are you mentioning it?

8. Line 347 - you can remove the second half of the sentence. It does not tie in well with the first part of the sentence.

Author Response

line 105 - a much more patchy....

Reply: change has been made (it is line 100 in the current version).

lines 138 - "Moreover, quantitative coronary angiography was used to assess the extent of coronary disease which is not a sensitive method to quantitatively estimate the degree of coronary calcification."Better to say "However the method used to quantify coronary calcification was not accurate."

Reply:  we changed the sentence (line 134-135) but we left the word ‘sensitive’. The reason is that we want to express that the method, even though it may pick changes in calcification, is not so sensitive. A method can be accurate in detecting changes but perhaps onlt above a certain threshold. However, if the reviewer or the editor insists that we use the word ‘accurate’, we will comply.

Line 150 "there was a positive effect..." You want to say 'nifedipine lead to regression of …".

Reply:  change has been made (line 143).

Lines 209 ff - I do not understand the logic here. Rewrite.

Reply:  Since line numbers in the version that we have to work on differ slightly from the version that the reviewer has had, we have to guess where exactly the logic went wrong. We presume it starts at line 206. We have rewritten these lines (207-209).

Lines 218ff - how can a non-calcified area reflect calcification enlargement? This does not make sense.

Reply:  What we meant to say is that calcification can occur in plaques that were non-calcified before treatment. We have adapted the text (line 217-218).

Lines 275-279 - the first half and second half of the sentence do not connect well. Rewrite.

Reply:  We presume it is the sentence in line 279-283 in the current file. We adapted the sentence.

Line 299-300 - since the study does not mention calcification, why are you mentioning it?

Reply:  This study assessed pulse wave velocity as a proxy of calcification; we have clarified this (line 302-303).

Line 347 - you can remove the second half of the sentence. It does not tie in well with the first part of the sentence.

Reply:  Assuming that this is line 352 in our version, we have deleted the second part of the sentence.